# Octogenarians’ Breast Cancer Is Associated with an Unfavorable Tumor Immune Microenvironment and Worse Disease-Free Survival

**DOI:** 10.3390/cancers13122933

**Published:** 2021-06-11

**Authors:** Maiko Okano, Masanori Oshi, Swagoto Mukhopadhyay, Qianya Qi, Li Yan, Itaru Endo, Toru Ohtake, Kazuaki Takabe

**Affiliations:** 1Breast Surgery, Department of Surgical Oncology, Roswell Park Comprehensive Cancer Center, Buffalo, NY 14263, USA; omaiko@hoshipital.jp (M.O.); masanori.oshi@roswellpark.org (M.O.); swagoto.mukhopadhyay@roswellpark.org (S.M.); 2Department of Breast Surgery, School of Medicine, Fukushima Medical University, Fukushima 960-1295, Japan; trcyn@fmu.ac.jp; 3Department of Gastroenterological Surgery, Graduate School of Medicine, Yokohama City University, Yokohama 236-0004, Japan; endoit@yokohama-cu.ac.jp; 4Department of Biostatistics & Bioinformatics, Roswell Park Comprehensive Cancer Center, Buffalo, NY 14263, USA; qianya.qi@roswellpark.org (Q.Q.); li.yan@roswellpark.org (L.Y.); 5Division of Digestive and General Surgery, Niigata University Graduate School of Medical and Dental Sciences, Niigata 951-8510, Japan; 6Department of Breast Surgery and Oncology, Tokyo Medical University, Tokyo 160-8402, Japan; 7Department of Surgery, Jacobs School of Medicine and Biomedical Sciences, University at Buffalo, The State University of New York, Buffalo, NY 14263, USA

**Keywords:** breast cancer, elderly, octogenarian, GSEA, CIBERSORT, tumor microenvironment, macrophage, geriatric oncology

## Abstract

**Simple Summary:**

In recent years, as the elderly population has grown, the number of elderly breast cancer patients has increased, but their biological characteristics are still controversial. This study investigated octogenarians’ breast cancer biology and its tumor microenvironment utilizing an in-silico translational approach to multiple large patient cohorts. We found that octogenarians’ breast cancer was associated with worse survival and an unfavorable tumor immune microenvironment such as M2 macrophage but not with aggressive cancer cell biology. Our report is important for understanding the characteristics of elderly breast cancer patients and would be critical for the development of breast cancer treatment in the future.

**Abstract:**

Elderly patients are known to have a worse prognosis for breast cancer. This is commonly blamed on their medical comorbidities and access to care. However, in addition to these social issues, we hypothesized that the extreme elderly (octogenarians—patients over 80 years old) have biologically worse cancer with unfavorable tumor immune microenvironment. The Cancer Genomic Atlas (TCGA) and the Molecular Taxonomy of Breast Cancer International Consortium (METABRIC) breast cancer cohorts were analyzed. The control (aged 40–65) and octogenarians numbered 668 and 53 in TCGA and 979 and 118 in METABRIC, respectively. Octogenarians had significantly worse breast cancer-specific survival in both cohorts (*p* < 0.01). Octogenarians had a higher ER-positive subtype rate than controls in both cohorts. Regarding PAM50 classification, luminal-A and -B subtypes were significantly higher in octogenarians, whereas basal and claudin-low subtypes were significantly lower (*p* < 0.05) in octogenarians. There was no difference in tumor mutation load, intratumor heterogeneity, or cytolytic activity by age. However, the octogenarian cohort was significantly associated with high infiltration of pro-cancer immune cells, M2 macrophage, and regulatory T cells in both cohorts (*p* < 0.05). Our results demonstrate that octogenarians’ breast cancer is associated with worse survival and with an unfavorable tumor immune microenvironment.

## 1. Introduction

The aging population is an increasingly central health and social issue in many developed countries—the health and quality of life of the elderly present unforeseen challenges to health systems. The risk of breast cancer increases with age, with approximately 20% of women diagnosed with breast cancer being over the age of 80 in the US [1]. Breast cancer screening and treatment have made remarkable progress in recent decades, and survival rates from breast cancer have improved remarkably throughout the world [2,3,4]. However, the outcomes in elderly breast cancer patients have not improved proportionally to the progress seen in younger patients [5]. There is speculation that octogenarians (patients greater than 80 years of age) may lack access to care or their comorbidities and frailty may prevent them from achieving similar progress.

Some reports indicate that, even when elderly patients have breast cancer with favorable characteristics, they suffer worse prognoses than younger patients [6]. Being aged 60 and older was demonstrated as an independent negative prognostic factor [6]. Furthermore, older age was associated with early mortality in the metastatic setting [7]. Conflictingly, some have reported that early-stage breast cancer in octogenarian women has similar breast cancer characteristics to younger women [8]. In fact, Muss et al. reported age is not an independent prognostic factor for outcomes [9]. Thus, it is difficult to conclude that medical comorbidities and worse access to care are primarily why elderly breast cancer has worse survival with such contradicting evidence. Further assessments for an underlying biological etiology are necessary to address the gap in age-based outcomes for breast cancer.

With advances in technology, it is now possible to study cancer biology utilizing comprehensive transcriptomes. Analysis using intrinsic subtype (e.g., luminal, basal-like, and HER-2 like) has become routine [10,11], and gene expression profiling is now common [12]. Furthermore, improved transcriptome and gene sequencing technologies have deepened our understanding of the tumor microenvironment [13,14,15]. Our group has recently elucidated the clinical relevance of stromal cells [16,17,18,19,20] and infiltrating immune cells [21,22,23,24,25,26,27,28,29,30,31] in the tumor microenvironment utilizing an in silico translational approach to multiple large patient cohorts. Using these various modalities, it has become clear that breast tumors differ at the DNA level depending on the age group [32].

This study hypothesized that the extreme elderly, octogenarians, have biologically worse cancers determined by the tumor immune microenvironment.

## 2. Materials and Methods

### 2.1. The Data of Breast Cancer Patient Cohorts

Two completely independent breast cancer cohorts, The Cancer Genome Atlas (TCGA) and the Molecular Taxonomy of Breast Cancer International Consortium (METABRIC), were used in the current study. It is essential to analyze cohorts with a reasonable number of octogenarian breast cancer patients to obtain statistically meaningful results, and these were the largest cohorts that we had access to. TCGA is a joint collaboration project of the National Cancer Institute (NCI) and the National Human Genome Research Institute of the National Institute of Health that includes 1084 breast cancer patients [33]. The METABRIC cohort was collected from tumor banks in the UK and Canada and includes 2509 breast cancer patients [34]. Both cohorts consist of treatment naïve primary cancer samples with clinical profiles, survival data, and gene expression data. They are de-identified publicly available databases; thus, Institutional Review Board (IRB) approval was waived. Our group has used both of these databases previously to demonstrate and unravel the tumor biology of breast cancer [16,17,18,19,20,21,22,23,24,25,26,27,28,29,30,31,35,36,37,38,39,40,41,42,43,44,45,46,47,48,49].

### 2.2. Statistical Analysis

In order to compare the biology between elderly and the most common age breast cancer patients, this study defined the control group as patients aged 40–65 and octogenarians (age over 80) at the time of breast cancer diagnosis. We used the Kaplan–Meier method with log-rank testing. Both the TCGA and the METABRIC cohorts provide the information on specifically breast cancer as the cause of death. To this end, the survival curve obtained from this information is Breast Cancer-Specific Survival (BCSS) following DATECAN guidelines [50]. Overall Survival (OS) indicates death by all causes. Fisher’s exact tests and one-way ANOVA tests were used to compare the differences between groups. In all analyses, *p*-values less than 0.05 were considered statistically significant. All statistical analyses were performed using Microsoft Excel 2010, R software (http:///www.r-project.org/, accessed on 1 April 2019), and Bioconductor (http://bioconductor.org/, accessed on 1 April 2019).

### 2.3. Gene Set Enrichment Analysis (GSEA)

Gene Set Enrichment Analysis (GSEA) was conducted to investigate what biology is associated with octogenarians’ breast cancer compared to common age. GSEA was calculated by the Broad Institute software (http://software.broadinstitute.org/gsea/index.jsp, accessed on 1 April 2019) [51], as previously described [35,36]. As recommended by GSEA software, a False Positive Rate (FDR) of 0.25 was defined as statistically significant.

### 2.4. The Immune Cytolytic Activity (CYT) Score, the Mutant-Allele Tumor Heterogeneity (MATH), and CIBERSORT

To substantiate our hypothesis that octogenarians’ breast cancer has worse biology, the following three analyses were performed. The immune Cytolytic Activity (CYT) score is a new indicator of cancer immunity developed by Rooney et al. [13]. It is acquired by calculating mRNA expression levels of Granzyme A (GZMA) and Perforin (PRF1) in the tumor and used to predict the level of immune-mediated elimination associated with Cytotoxic T Cell (CTL) markers. We previously demonstrated that CYT score reflects immune cell antitumor activity and its clinical relevance in several articles [17,18,22,24,26,27,28,29,31,35,36,37,41,43,44,46,47,48,49,52,53,54].

Intratumor heterogeneity is the clonal variation between individual cancer cells within a patient’s tumor. It has recently drawn attention as an important underlying mechanism of therapeutic failure. Our group utilized Mutant Allele Tumor Heterogeneity (MATH) out of the multiple computational algorithms that assess intratumor heterogeneity based on the exome sequencing of tumors [14]. Our prior work assessed intratumor heterogeneity and demonstrated that it is associated with cancer biology and patient outcomes [30,36,37,41,42,55].

Recently, the tumor immune microenvironment has been found to plays an important role in cancer biology, and the profiling of the infiltrating immune cells is critical to assess the characteristics of a tumor [56]. CIBERSORT deconvolution algorithm was utilized by Newman et al. to estimate tumor-infiltrating immune cells in the tumor microenvironment using gene expression profiles [15]. The 22 cell fractions were calculated via their online calculator (https://cibersort.stanford.edu/, 1 April 2019). We have already illustrated that CIBERSORT analysis could provide the clinical significance and the prediction of outcomes in some cancers in several articles [21,27,28,29,30,31,35,36,41,48,52,57]. 

## 3. Results

### 3.1. Octogenarians Have a Significantly Higher Rate of Luminal Type Cancers Than the Control Group in Both Cohorts

We assessed the trend of luminal subtype breast cancers in our data to see if our cohorts followed the known tendency that breast cancer in the elderly has a higher ratio of luminal subtypes compared than younger patients. In total, 1081 patients with age information were available in TCGA and 1866 in METABRIC. These cohorts were separated into the control and the octogenarian groups, numbering 668 and 53 in TCGA and 979 and 118 in METABRIC, respectively. We examined clinical factors for each patient group and the distribution of breast cancer subtypes in octogenarians’ breast cancer (Table 1). The American Joint Committee on Cancer (AJCC) pathological stage, histology, and Nottingham grade had no statistical difference between the age groups in either of the cohorts. Estrogen Receptor (ER) expression determined by immunohistochemistry was significantly higher in the octogenarian group than in the control group in both the METABRIC and the TCGA cohorts. However, there was no difference in Progesterone Receptor (PR) expression between the age groups in either cohort (Figure 1). Assessment of the intrinsic subtypes by gene expression showed that Luminal-A and -B subtypes were significantly higher, and basal and claudin-low subtypes were significantly lower in octogenarians in both cohorts, analogous to previous reports (Figure 1, both *p* < 0.013).

### 3.2. Octogenarians Have Worse Overall Survival and Higher Recurrence Rates in Two Independent Cohorts

By the very nature of aging, we could likely predict that octogenarians have worse OS compared to younger patients; however, to address our hypothesis, we assessed for any difference in Breast Cancer Specific Survival (BCSS) to reflect the aggressiveness of breast cancer in each age group. The ER-positive subtype was of particular interest since the octogenarians appeared to have that subtype more commonly.

Octogenarians demonstrated significantly worse BCSS overall and in ER-positive subtypes in both the TCGA and the METABRIC cohorts (Figure 2, all *p* < 0.015). As expected, the OS Kaplan–Meier curve of the octogenarians was significantly worse than the control group in both the whole cohorts and ER-positive subtypes in TCGA (Figure 2, both *p* < 0.001). These results show that octogenarians have worse outcomes from breast cancer than the control group, which could be due to worse biology or differences in management.

### 3.3. There Is No Significant Difference in Tumor Mutation Load, Intratumor Heterogeneity, or Cytolytic Activity between the Age Groups

Given that octogenarians demonstrated poorer survival within the whole cohorts and in the ER-positive subtypes, we expected them to have worse cancer biology, which could be reflected in a higher mutation load, intratumor heterogeneity by Mutant Allele Tumor Heterogeneity (MATH), or Cytolytic Activity Score (CYT). Surprisingly, there was no difference between the age groups in mutation load, intratumor heterogeneity, or cytolytic activity (Figure 3).

### 3.4. The Octogenarian Group Was Not Associated with Any Gene Set Related to Cancer Aggressiveness by Gene Set Enrichment Analysis (GSEA) in Both the TCGA and the METABRIC Cohorts

GSEA of the Hallmark collection was conducted to investigate any specific pathways associated with octogenarians’ breast cancer. Unexpectedly, octogenarians’ breast cancer was not associated with any gene set associated with cancer aggressiveness, such as angiogenesis, epithelial–mesenchymal transition, KRAS signaling pathway, PI3K-AKT signaling pathway, MYC targets, or mitotic spindle in either the TCGA or the METABRIC cohort (Figure 4, showing only results for TCGA).

### 3.5. Octogenarians’ Tumors Had High Infiltration of Pro-Cancer Immune Cells M2 Macrophage and Regulatory T Cells While Having Lower Infiltration of Anti-Cancer Immune Cells M1 Macrophage and Activated Memory CD4 T Cells

Given the GSEA result that octogenarians did not enrich tumor aggressiveness pathways, we investigated the tumor immune microenvironment of octogenarians’ breast cancers to explain the poorer outcomes. We performed CIBERSORT analysis to estimate the immune cell composition infiltrating the tumors. We found that infiltration of pro-cancer immune cells, the M2 macrophage, was significantly higher in the octogenarian group across both the TCGA and the METABRIC cohorts (Figure 5, both *p* < 0.001). Additionally, regulatory T cells, another pro-cancer immune cell, also highly infiltrated octogenarians’ breast cancers in TCGA. There was a similar trend in METABRIC (*p* = 0.023 and *p* = 0.054, respectively). Conversely, anti-cancer immune cells, M1 macrophages, and activated memory CD4 T cells both had lower infiltration in octogenarians in the METABRIC cohort (both *p* < 0.05). Together with the survival outcomes, mutation data, and GSEA results, we concluded that octogenarians’ breast cancer was associated with worse survival than younger patients. Our findings suggest that this is because octogenarians’ breast cancers may not inherently have more aggressive cancer biology, but it may have a worse tumor immune microenvironment.

### 3.6. Analyses by Subtype

Given the fact that cancer biology differs significantly by subtype, survival, mutation load, intratumor heterogeneity, cytolytic activity, and immune cell infiltration were individually assessed by the subtype. As shown in Appendix A, octogenarians demonstrated significantly worse BCSS only in Luminal-A. Octogenarians showed no difference in mutational load, intratumor heterogeneity, or cytolytic activity in any of the subtypes except mutational load in Her2 subtype (Appendix A). No significant infiltration in any macrophages, CD4, CD8, or Regulatory T cells were seen in octogenarians in any subtypes (Appendix A).

## 4. Discussion

We found that the octogenarian cohort had more luminal type breast cancer, which is known to have a relatively good prognosis. However, their overall survival and, perhaps more importantly, their disease-free and disease-specific survival were worse than the younger control group. We found no difference in cancer cell biology, such as tumor mutation load, intratumor heterogeneity, cytolytic activity, and tumor aggravating pathways of cancer cells and stromal cells. However, we found that octogenarians’ breast cancer had a significantly higher infiltration of pro-cancer immune cells, M2 macrophage, and regulatory T cells and a lower infiltration of anti-cancer immune cells, M1 macrophage, and activated memory CD4 T cells compared to younger controls. We concluded that octogenarians’ breast cancer was associated with worse survival, possibly explained by an unfavorable tumor immune microenvironment but not necessarily with a more aggressive cancer cell biology.

M2 macrophages have anti-inflammatory properties and are found in the tumor development phase [58,59], promote angiogenesis in vivo [60], and correlate with poor prognosis in breast cancer [61,62]. Multiple studies show increased M2 macrophage infiltration in elderly patients’ bone marrow, lymph nodes, spleen, lung, and muscles [63,64,65,66]. M2 macrophages release cytokines and suppress cytotoxic lymphocytes through PD-L1 expression and recruitment of Tregs [67]. Our results align with previous reports that tumors with an increased presence of M2 macrophages have a poorer prognosis and that octogenarians have more M2 macrophages than younger patients. Furthermore, an increased number of Tregs is associated with immune deficiency in aged animals [68] and promotes tumor growth. Based upon our results, we speculate that the prognosis of octogenarians’ breast cancer is worse than breast cancer in the young not only because of inadequate access to care or increased medical comorbidities but also because of an unfavorable tumor immune microenvironment.

In clinical practice, octogenarians are more likely to receive nonstandard care or be undertreated [69,70,71] because of their medical comorbidities, frailty, concern for treatment toxicity, or social determinants of health. Often performance status and comorbidities rather than cancer biology dictate the management strategy in elderly patients [72]. About half of early-stage breast cancer patients over 70 years old were reported to have sarcopenia [73]. Further, data on the effect and toxicity of newer drugs are often limited for elderly patients since patients over 70 years old are typically excluded from clinical trials [74]. Although the importance of comprehensively evaluating elderly patients before any treatment cannot be overemphasized, there is evidence that elderly patients tolerate and benefit from standard therapy [75]. However, the cited cohort includes elderly patients under 80 years old. For example, a surgical approach with hormone therapy also clearly improves survival in elderly breast cancer patients with early-stage hormone-receptor-positive disease compared to hormone therapy alone [76].

It should be noted that the elderly are more likely to have suboptimal adherence to hormone therapy, which can increase the risk of cancer recurrence [77,78]. Our findings show that the tumor microenvironment of the octogenarian is significantly different and worse than younger patients. Some studies report that patient immune profiles in elderly breast cancer patients are changed with chemotherapy, and these profiles can predict the clinical frailty of patients [79]. The need for such studies is paramount given the rapid aging of society with increasing longevity. In fact, a combined approach investigating appropriate therapies for elderly patients with breast cancer, along with standardized assessments of frailty and comorbidities, is in urgent need. This would allow for evidence-based treatment decisions founded upon a holistic assessment of the patient, including their tumor biology, treatment goals, life expectancy, and physical and mental condition.

This study is not free of limitations. The largest limitation is that, although we retrospectively demonstrated the association between octogenarians’ breast cancer and the infiltration of immune cells, the presented data do not show causality. Another limitation is that all the patients included in the TCGA and the METABRIC cohorts underwent surgical resection of their tumor in order to be included in these databases. To this end, breast cancer treated exclusively with endocrine therapy, which can be the case especially in older women, may have been excluded, resulting in a selection bias. In addition, some key clinical information that would have been helpful for this study, such as comorbidities or geriatric characteristics, were missing in these cohorts. In addition, it would have been ideal to group the patients by menopausal status; however, that information was lacking in both of the analyzed cohorts, and the sample size would be too small for meaningful analyses if we limited the control group to those aged 55–65. Menopause significantly changes the internal environment of women and causes various symptoms [80]. It is well known that menopause by itself has a major impact on breast cancer biology. Medication for hormone-sensitive breast cancer differs in premenopausal and postmenopausal patients [81]. It has also been reported that zoledronic acid has different therapeutic effects before and after menopause [82]. However, few studies have compared the cancer biology by the menstruation status instead of age. To this end, it would be interesting to investigate the difference among premenopausal, postmenopausal under 70 years old, and older patients, since we were unable to do this due to the lack of menstruation information in our cohorts. By comparing the biology of pre- and postmenopausal cancer at the genetic level and clarifying the characteristics of each, it could be possible to contribute to the prevention and treatment of breast cancer.

## 5. Conclusions

Our results demonstrate that octogenarians’ breast cancer is associated with an unfavorable tumor immune microenvironment that may explain its biologically worse behavior.

## Figures and Tables

**Figure 1 cancers-13-02933-f001:**
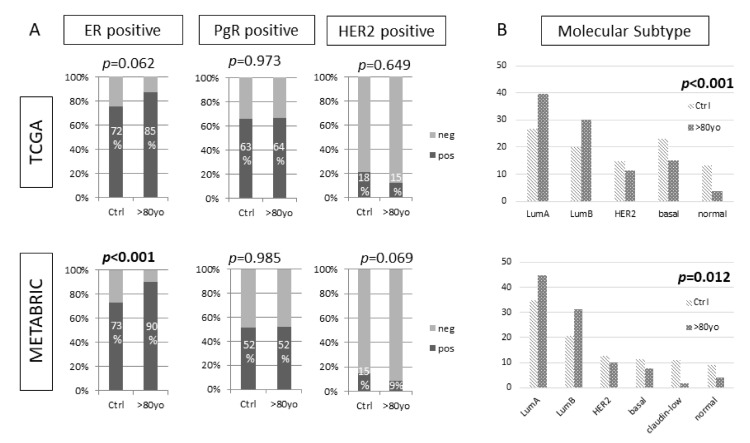
Expression rate of receptors and intrinsic subtype in TCGA and the METABRIC cohorts. (**A**) Comparison of the positive expression rate of receptors in the control group versus the octogenarian group. (**B**) Comparison of the PAM50 intrinsic subtypes by gene expression in the control group versus the octogenarian group. There was a statistically significant difference between the control group and the octogenarians.

**Figure 2 cancers-13-02933-f002:**
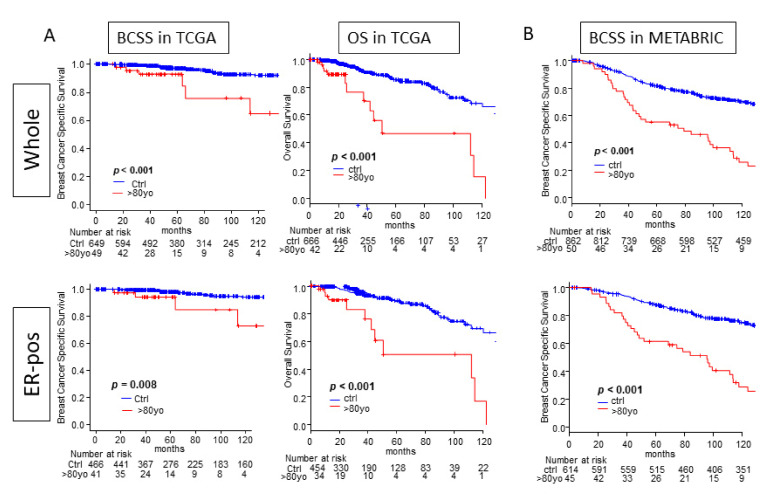
Survival analyses. (**A**) Kaplan–Meier survival curves for breast cancer specific survival (BCSS) and overall survival (OS) between the octogenarian group and the control group in the TCGA cohort. (**B**) Kaplan–Meier survival curves for BCSS between the octogenarian group and the control group in the METABRIC cohort.

**Figure 3 cancers-13-02933-f003:**
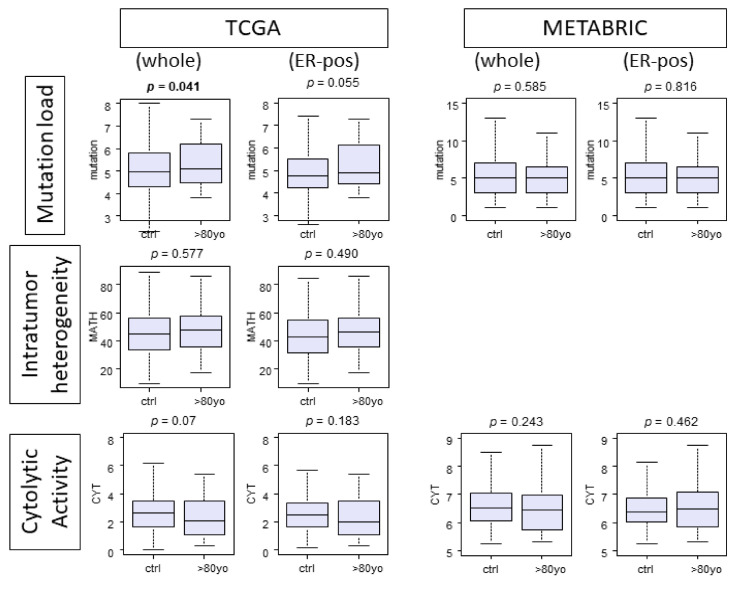
Boxplots comparing the control group and the octogenarians by mutation load, Mutant-Allele Tumor Heterogeneity (MATH) score, and Cytolytic Activity Score (CYT). The analysis was done with the whole cohorts and ER positive groups.

**Figure 4 cancers-13-02933-f004:**
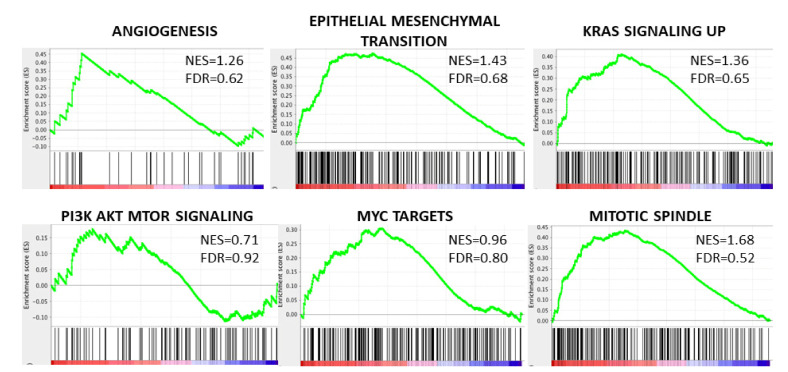
Gene Set Enrichment Analysis (GSEA) of the octogenarian group with Normalized Enrichment Score (NES) and False Discovery Rate (FDR). Representative gene sets associated with cancer malignancy in GSEA analysis (TCGA shown).

**Figure 5 cancers-13-02933-f005:**
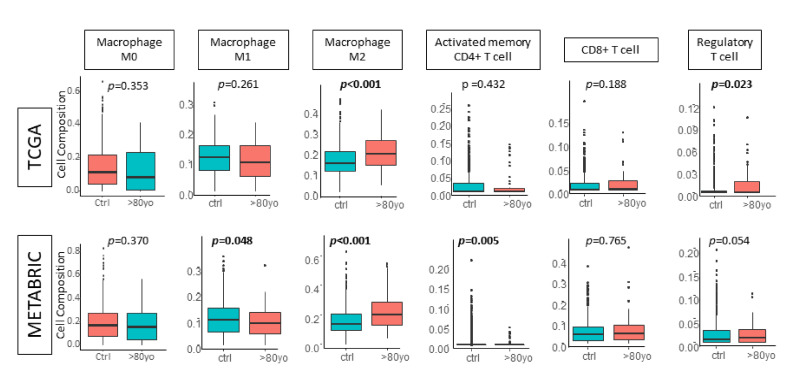
The CIBERSORT algorithm in the TCGA and the METABRIC cohorts. Boxplots comparing the control group and the octogenarian group by macrophage M0, macrophage M1, macrophage M2, activated memory CD4+ T cell, CD8+ T cell, and regulatory T cells.

**Table 1 cancers-13-02933-t001:** Clinical factors of the each patient group.

TCGA								METABRIC							
	Age Group								Age Group						
Variable	Total		Control (40>, ≤60 years)		Octogenarian (>80 years)		*p*-Value	Variable	Total		Control (40>, ≤60 years)		Octogenarian (>80 years)		*p*-Value
	N	%	N	%	N	%			N	%	N	%	N	%	
Total	1081	100	668	100	53	100		Total	1866	100	979	100	118	100	
pTMN stage								T stage							
I	181	16.7	108	16.2	11	20.8	0.113	1	464	24.9	298	30.4	10	8.5	0.135
II	611	56.5	391	58.5	22	41.5		2	777	41.6	391	39.9	62	52.5	
III	246	22.8	140	21	17	32.1		3	113	6.1	49	5	10	8.5	
IV	19	1.8	14	2.1	1	1.9		4	8	0.4	3	0.3	1	0.8	
Unknown	24	2.2	15	2.2	2	3.8		Unknown	504	27	238	24.3	35	29.7	
Estrogen receptor								Estrogen receptor							
positive	794	73.5	478	71.6	45	84.9	0.062	positive	1431	76.7	714	72.9	106	89.8	0
negative	237	21.9	159	23.8	7	13.2		negative	435	23.3	265	27.1	12	10.2	
Unknown	50	4.6	31	4.6	1	1.9		Unknown	0	0	0	0	0	0	
Progesterone receptor								Progesterone receptor							
positive	688	63.6	418	62.6	34	64.2	0.973	positive	993	53.2	507	51.8	61	51.7	0.985
negative	340	31.5	219	32.8	18	34		negative	873	46.8	472	48.2	57	48.3	
Unknown	53	4.9	31	4.6	1	1.9		Unknown	0	0	0	0	0	0	
HER2								HER2							
positive	179	16.6	109	16.3	8	15.1	0.649	positive	230	12.3	143	14.6	10	8.5	0.069
negative	762	70.5	477	71.4	42	79.2		negative	1636	87.7	836	85.4	108	91.5	
Unknown	140	13	82	12.3	3	5.7		Unknown	0	0	0	0	0	0	
Histology								Histology							
Infiltrating Ductal	771	71.3	482	72.2	34	64.2	0.195	Infiltrating Ductal	1475	79	744	76	94	79.7	0.247
Infiltrating Lobular	203	18.8	130	19.5	13	24.5		Infiltrating Lobular	137	7.3	79	8.1	11	9.3	
Others	107	9.9	56	8.4	6	11.3		Others	252	13.5	156	15.9	13	11	
Histological Grade								Unknown	2	0.1	0	0	0	0	
1	77	7.1	52	7.8	4	7.5	0.088	Histological Grade							
2	268	24.8	170	25.4	15	28.3		1	159	8.5	99	10.1	11	9.3	0.073
3	232	21.5	154	23.1	12	22.6		2	723	38.7	356	36.4	57	48.3	
Unknown	450	41.6	258	38.6	20	37.7		3	912	48.9	488	49.8	43	36.4	
PAM50 subtype								Unknown	72	3.9	36	3.7	7	5.9	
Luminal A	303	28	179	26.8	21	39.6	0	claudin subtype							
Luminal B	244	22.6	133	19.9	16	30.2		Luminal A	673	36.1	343	35	53	44.9	0.012
HER2	144	13.3	98	14.7	6	11.3		Luminal B	454	24.3	203	20.7	37	31.4	
Basal-like	227	21	154	23.1	8	15.1		HER2	218	11.7	125	12.8	12	10.2	
normal-like	136	12.6	88	13.2	2	3.8		Basal-like	198	10.6	111	11.3	9	7.6	
Unknown	27	2.5	16	2.4	0	0		claudin-low	182	9.8	108	11	2	1.7	
Race								normal-like	135	7.2	88	9	5	4.2	
White	745	68.9	476	71.3	33	62.3	0.626	Unknown	6	0.3	1	0.1	0	0	
Black	180	16.7	108	16.2	6	11.3									
other	156	14.4	84	12.6	14	26.4									

## Data Availability

All data were from previous studies.

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
