# Peer review of "Octogenarians’ Breast Cancer Is Associated with an Unfavorable Tumor Immune Microenvironment and Worse Disease-Free Survival"

_cancers, 2021, doi:10.3390/cancers13122933_

Round 1

Reviewer 1 Report

You studied the prognosis of breast cancer in the elderly. I believe that this result is of interest for many physicians. Their prognosis is worse than that of young patients, and macrophages are involved. Focusing on and investigating the underlying causes is also an important finding in the future development of treatments for elderly breast cancer.

I'm not a statistician, so I don't understand the comparison of Kaplan-Meier curves in detail. It is natural for over all survival, but I would like to ask about the analysis method whether it can be compared properly with young breast cancer depending on whether DSS includes death from other diseases.

Author Response

Author’s Point-by-point Response to Reviewer #1

[cancers-1219016]

Octogenarian Breast Cancer is Associated with Unfavorable Tumor Immune Microenvironment and Worsened Disease-Free Survival

Reviewer1:

You studied the prognosis of breast cancer in the elderly. I believe that this result is of interest for many physicians. Their prognosis is worse than that of young patients, and macrophages are involved. Focusing on and investigating the underlying causes is also an important finding in the future development of treatments for elderly breast cancer.

Response:

We would like to thank Reviewer1 for taking the time to review and for providing constructive comments. We are delighted to learn that he/she felt that our result is of interest for many physicians.

Comment 1:

I'm not a statistician, so I don't understand the comparison of Kaplan-Meier curves in detail. It is natural for overall survival, but I would like to ask about the analysis method whether it can be compared properly with young breast cancer depending on whether DSS includes death from other diseases.

Response 1:

We agree with this comment, which is essentially the same as Editor’s Comment #1. Overall survival as a primary endpoint is inappropriate for geriatric population thus we shifted the focus to disease-related outcome. We have changed the figures and the description in the article as below.

“Figure 2. (A) Kaplan–Meier survival curves for breast cancer specific survival (BCSS) and overall survival (OS) between the Octogenarian group and the control group in TCGA cohort. (B) Kaplan–Meier survival curves for BCSS between the Octogenarian group and the control group in the METABRIC cohort.“

Materials and Methods 2.2. section:

We used the Kaplan–Meier method with log-rank testing to determine breast cancer-specific survival (BCSS) and overall survival (OS) following DATECAN guidelines [ref].

Result 3.2. section:

As a result of the analysis, Octogenarians demonstrated significantly worse breast cancer-specific survival (BCSS) overall and in ER-positive subtypes in both TCGA and the METABRIC cohorts (Figure 2, all p < 0.015). As expected, the OS Kaplan-Meier curve of the Octogenarians was significantly worse than the control group in both the whole cohorts and ER-positive subtypes in TCGA (Figure 2, both p < 0.001). These results show that Octogenarians have worse outcomes from breast cancer, which could be due to worse biology or differences in management than the control group.

Reviewer 2 Report

This paper is especially interesting because it is aimed at a population of the oldest old that has not usually been investigated and is worthy of treatment.

I would ask the authors to highlight for readers some more aspects regarding this population and its features regardless of the settings (ie Perspectives and limits of cancer treatment in an oldest old population.. doi: 10.1007/s40520-021-01821-2.) . Mainly the biomarker of aging and the age-related physiological changes that can impact treatments, those conditions such as sarcopenia (ie Muscoloskeletal aging, sarcopenia and cancer), for example, that are very frequent in this population or frailty in breast cancer patients (ie Different Impact of Definitions of Sarcopenia in Defining Frailty Status in a Population of Older Women with Early Breast Cancer). In the discussion,  the importance of evaluating these patients before deciding on treatment should be emphasized more; many papers in the literature underline the need for a correct assessment of the older cancer patients.

Author Response

Author’s Point-by-point Response to Reviewer #2

[cancers-1219016]

Octogenarian Breast Cancer is Associated with Unfavorable Tumor Immune Microenvironment and Worsened Disease-Free Survival

Reviewer2

Comment:

This paper is especially interesting because it is aimed at a population of the oldest old that has not usually been investigated and is worthy of treatment.

Response:

We would like to thank Reviewer2 for his/her thorough review of our manuscript. We are delighted to learn that the Reviewer found our paper to be interesting.

Comment 1:

I would ask the authors to highlight for readers some more aspects regarding this population and its features regardless of the settings (ie Perspectives and limits of cancer treatment in an oldest old population. doi: 10.1007/s40520-021-01821-2.). Mainly the biomarker of aging and the age-related physiological changes that can impact treatments, those conditions such as sarcopenia (ie Musculoskeletal aging, sarcopenia and cancer), for example, that are very frequent in this population or frailty in breast cancer patients (ie Different Impact of Definitions of Sarcopenia in Defining Frailty Status in a Population of Older Women with Early Breast Cancer). In the discussion, the importance of evaluating these patients before deciding on treatment should be emphasized more; many papers in the literature underline the need for a correct assessment of the older cancer patients.

Response 1:

We agree with the Reviewer that geriatric patients have unique challenges and appreciate their highlighting this important aspect. We have emphasized the importance of evaluating these patients prior to treatment and added sentences in the discussion section as below.

Discussion section:

Often performance status and comorbidities rather than cancer biology dictate the management strategy in elderly patients [ref]. About half of early-stage breast cancer patients over 70 years old were reported to have sarcopenia [ref]. Further, data on the effect and toxicity of newer drugs are often limited for elderly patients since patients over 70 years old are typically excluded from clinical trials [ref]. Although the importance of comprehensively evaluating elderly patients before any treatment cannot be overemphasized, there is evidence that elderly patients tolerate and benefit from standard therapy. However, the cited cohort includes elderly patients under 80 years old. For example, a surgical approach with hormone therapy also clearly improves survival in elderly breast cancer patients with early-stage hormone-receptor-positive disease compared to hormone therapy alone. It should be noted that the elderly are more likely to have suboptimal adherence to hormone therapy which can increase the risk of cancer recurrence. Our findings show that the tumor immune microenvironment of the octogenarian is significantly different from younger patients. Some studies report that patient immune profiles in elderly breast cancer patients are changed with chemotherapy, and these profiles can predict the clinical frailty of patients [ref]. The need for such studies is paramount given the rapid aging of society with increasing longevity. In fact, a combined approach investigating appropriate therapies for elderly patients with breast cancer, along with standardized assessments of frailty and comorbidities, is in urgent need. This would allow for evidence-based treatment decisions founded upon a holistic assessment of the patient, including their tumor biology, treatment goals, life expectancy, and physical and mental condition.

Reviewer 3 Report

Relevant and interesting topic but major concerns about the manuscript

- the choice to take 40-65y for the young cohort is far from optimal. About 1/3 will be premenopausal and 2/3 menopausal. It is well known that there may be huge differences in microenvironment depending on menopause. It would be much better to take eg a group 30-45, and a group 55-65, as control.

- major methodological issue is that they don't correct for the differences in molecular subtype. It would have been much more logical to take a group of luminal A cases in each age group, and then evaluate the markers they investigate; and the same for the other subtypes.

- DFS = distant relapse free survival. this is very confusing! Please refer to the Datecan definition of endpoints (ann oncol). This should be called DRFS.  DSS is also a bit confusing. I suppose they mean breast cancer specific survival (BCSS) but this implies that they know the cause of death from all patients, and exclude the patients that died from another cause (which is frequent in breast cancer! Certainly in elderly)

- concepts of 'cytolytic activity', and 'intratumor heterogeneity' are very poorly explained, and not understandable/interpretable by persons not familiar with these terms.

Minor:

- quality of English writing is suboptimal

Author Response

Author’s Point-by-point Response to Reviewer #3

[cancers-1219016]

Octogenarian Breast Cancer is Associated with Unfavorable Tumor Immune Microenvironment and Worsened Disease-Free Survival

Reviewer3

Relevant and interesting topic but major concerns about the manuscript

Response:

We would like to thank Reviewer #3 for his/her thoughtful and thorough review of our manuscript. We are delighted to learn that the Reviewer found our study to be relevant and interesting topic. Please find our point-by-point responses below.

Comment 1:

the choice to take 40-65y for the young cohort is far from optimal. About 1/3 will be premenopausal and 2/3 menopausal. It is well known that there may be huge differences in microenvironment depending on menopause. It would be much better to take eg a group 30-45, and a group 55-65, as control.

Response 1:

We agree the reviewer3 that it would be ideal to set the control group base upon menopausal status. There are two reasons we were unable to do so. First, the two cohorts we used lack the information on menopause. Second, the sample sizes were too small to have meaningful statistical analyses when the controls were set by the age 55-65. We believe this is a limitation of our analyses, thus added below sentence to the Discussion section.

Discussion section:

Also, we would have preferred to base the control group on menopausal status; however, that information was lacking in the two cohorts available, and the sample size would be too small to assess for true differences if we limited the control to age 55-65.

Comment 2:

major methodological issue is that they don't correct for the differences in molecular subtype. It would have been much more logical to take a group of luminal A cases in each age group, and then evaluate the markers they investigate; and the same for the other subtypes.

Response 2:

We agree with the Reviewer that the analyses could be more logical to perform the analysis by each subtype. The analyses by subtype were added in the Supplemental figures as below.

Result 3.6. section:

Given the fact that cancer biology differs significantly by subtype, survival, mutation load, intratumor heterogeneity, cytolytic activity and immune cell infiltration were individually assessed by the subtype. As shown in Supplemental Figure S1, octogenarian demonstrated significantly worse BCSS only in Luminal A. Octogenarian shown no difference in mutational load, intratumor heterogeneity, nor cytolytic activity in any of the subtypes except mutational load in Her2 subtype (Supplemental Figure S2). No significant infiltration in any macrophages, CD4, CD8, nor Regulatory T cells were seen in octogenarian in any subtypes (Supplemental Figure S3).

Supplemental Figure S1: Kaplan–Meier survival curves of breast cancer specific survival (BCSS) and overall survival (OS) analyses between the Octogenarian (red line) and the control group (blue line) by subtypes in TCGA cohort.

Supplemental Figure S2: Boxplots of comparison of the control group and the Octogenarians in mutation load, Intratumor heterogeneity (MATH score), and Cytolytic Activity Score (CYT) by the subtypes in TCGA cohort.

Supplemental Figure S3: Fraction of Macrophage (M0), M1, M2 activated memory CD4 T cell, CD8 T cell and Regulatory T cell in control group and Octogenarian by the subtypes in TCGA cohort.

Comment 3:

DFS = distant relapse free survival. this is very confusing! Please refer to the Datecan definition of endpoints (ann oncol). This should be called DRFS.  DSS is also a bit confusing. I suppose they mean breast cancer specific survival (BCSS) but this implies that they know the cause of death from all patients, and exclude the patients that died from another cause (which is frequent in breast cancer! Certainly in elderly)

Response 3:

DFS was a typographical error. We apologize for the confusions that it may have caused. We agree that DSS should be BCSS and amended throughout the manuscript including the description in the Results section and corrected the Figure 1 legend as outlined below.

“Figure 2. (A) Kaplan–Meier survival curves for breast cancer specific survival (BCSS) and overall survival (OS) between the Octogenarian group and the control group in TCGA cohort. (B) Kaplan–Meier survival curves for BCSS between the Octogenarian group and the control group in the METABRIC cohort. “

Materials and Methods 2.2. section:

We used the Kaplan–Meier method with log-rank testing to determine breast cancer-specific survival (BCSS) and overall survival (OS) following DATECAN guidelines [ref]. 

Result 3.2. section:

As a result of the analysis, Octogenarians demonstrated significantly worse breast cancer-specific survival (BCSS) overall and in ER-positive subtypes in both TCGA and the METABRIC cohorts (Figure 2, all p < 0.015). As expected, the OS Kaplan-Meier curve of the Octogenarians was significantly worse than the control group in both the whole cohorts and ER-positive subtypes in TCGA (Figure 2, both p < 0.001). These results show that Octogenarians have worse outcomes from breast cancer, which could be due to worse biology or differences in management than the control group.

Comment 4:

concepts of 'cytolytic activity', and 'intratumor heterogeneity' are very poorly explained, and not understandable/interpretable by persons not familiar with these terms.

Response 4:

We agree with the Reviewer that the explanations on cytolytic activity and intratumor heterogeneity were rather poor. We have added some descriptions in the Method section as below.

Materials and Methods 2.4. section:

The immune cytolytic activity (CYT) score is a new indicator of cancer immunity developed by Rooney et al [ref]. It is acquired by calculating mRNA expression levels of granzyme A (GZMA) and perforin (PRF1) in the bulk tumor and used to predict immune-mediated elimination level that associated with cytotoxic T cell (CTL) markers. We have previously demonstrated that CYT score reflects immune cell killing activity and its clinical relevance in several articles [ref]. Intratumor heterogeneity is the clonal variation between individual cancer cells within a patient's tumor. It has recently drawn attention as an important underlying mechanism of therapeutic failure. Our group utilizes Mutant Allele Tumor Heterogeneity (MATH) out of the multiple computational algorithms that assess intratumor heterogeneity based on the exome sequencing of tumors [ref]. Our prior work assessed intratumor heterogeneity and demonstrated that it is associated with cancer biology and patient outcomes [ref].

 Comment 5:

quality of English writing is suboptimal

Response 5:

We agree with the Reviewer that the quality of English writing can be improved. One of the authors who is a native English speaker with a graduate degree, Swagoto Mukhopadhyay, thoroughly edited the entire manuscript.

Round 2

Reviewer 2 Report

The changes made by the authors make the paper complete from a clinical point of view and very interesting and undoubtedly worth citing, considering the little-studied topic.

Author Response

[cancers-1219016]

Octogenarian Breast Cancer is Associated with Unfavorable Tumor Immune Microenvironment and Worsened Disease-Free Survival

Reviewer2

Comment:

The changes made by the authors make the paper complete from a clinical point of view and very interesting and undoubtedly worth citing, considering the little-studied topic.

Response:

We would like to sincerely thank Reviewer #2 for taking his/her time and effort reviewing our manuscript. We are delighted to learn that the Reviewer found our changes made the paper complete and very interesting.

Reviewer 3 Report

The manuscript has clearly improved. Some more issues need to be improved - They deleted distant relapse free survival which is ok. But they suddenly introduce breast cancer specific survival. where did they get the information about cause of death?? This is very difficult to obtain in general. This should be clarified. - the methods section should more clearly contain the research questions they had prior to starting the study. Now they just mention the tools they used - the last 4 lines of discussions need to be more extensive. They should explain that menopause by itself can have major impact on biology, and that it would have been interesting to investigate the differences between premenopausal, postmenopausal <70, and older pts

Author Response

[cancers-1219016]

Octogenarian Breast Cancer is Associated with Unfavorable Tumor Immune Microenvironment and Worsened Disease-Free Survival

 Reviewer3

Comment 1:

The manuscript has clearly improved. Some more issues need to be improved -

They deleted distant relapse free survival which is ok. But they suddenly introduce breast cancer specific survival. where did they get the information about cause of death?? This is very difficult to obtain in general. This should be clarified.

Response 1:

We would like to thank the Reviewer #3 for pointing out our mistakes. First, we were confused with the abbreviation of DFS, which is disease free survival. Second, we were not clear on cause of death when we demonstrated the survival curves. As matter of fact, both TCGA and METABRIC cohorts provide the information on specifically breast cancer as the cause of death. To this end, we demonstrate breast cancer specific survival (BCSS) and overall survival (OS) for TCGA and BCSS for METABRIC. We have added some explanation in Method section and renewed the figure.

Method section 2.2.:

We used the Kaplan–Meier method with log-rank testing. Both TCGA and METABRIC cohorts provide the information on specifically breast cancer as the cause of death. To this end, the survival curve obtained from this information are breast cancer specific survival (BCSS) following DATECAN guide-lines [50]. Overall survival (OS) demonstrate death by all causes.

Comment 2:

The methods section should more clearly contain the research questions they had prior to starting the study. Now they just mention the tools they used -

 Response 2:

We agree with the Reviewer that the method section should contain the research question more clearly. We added some sentences in the method section as below.

 2.1. Breast Cancer Patient Cohorts

Two completely independent breast cancer cohorts, The Cancer Genome Atlas (TCGA) and the Molecular Taxonomy of Breast Cancer International Consortium (METABRIC) were used in the current study. It is essential to analyze the cohorts with reasonable number of Octogenarian breast cancer patients to obtain statistically meaningful results, and these were the largest cohorts that we had access to. TCGA is a joint collaboration project of the National Cancer Institute (NCI) and the National Human Genome Research Institute of the National Institute of Health that includes 1084 breast cancer patients [33]. The METABRIC cohort was collected from tumor banks in the UK and Canada and includes 2509 breast cancer patients [34]. Both cohorts consist of treatment naïve primary cancer samples with clinical profiles, survival data, and gene expression data. They are de-identified publicly available databases; thus, institutional review board (IRB) approval was waived. Our group has used both of these databases previously to demonstrate and unravel the tumor biology of breast cancer [16-31,35-49].

 2.2. Statistical Analysis

In order to compare the biology between super-old and the most common age breast cancer patients, this study defined the control group as patients aged 40-65 and octogenarians (age over 80) at the time of breast cancer diagnosis. We used the Kaplan–Meier method with log-rank testing Two cohort has the information of disease specific survival, which is considered only breast cancer death, so we used the information as breast cancer specific survival following DATECAN guide-lines [50]. Fisher's exact tests and one-way ANOVA tests were used to compare the differences between groups. In all analyses, p-values less than 0.05 were considered statistically significant. All statistical analyses were performed using Microsoft Excel 2010, R software (http:///www.rproject.org/), and Bioconductor (http://bioconductor.org/).

 2.3. Gene Set Enrichment Analysis (GSEA)

Gene Set Enrichment Analysis (GSEA) was conducted to investigate what biology are associated with Octogenarian breast cancer compared to common age. GSEA was calculated by the Broad Institute software (http://software.broadinstitute.org/gsea/index.jsp) [51], as previously de-scribed [35,36]. As recommended by GSEA software, a False Positive Rate (FDR) of 0.25 was defined as statistically significant.

2.4. The Immune Cytolytic Activity (CYT) Score, the Mutant-Allele Tumor Heterogeneity (MATH), and CIBERSORT

To substantiate our hypothesis that Octogenarian breast cancer has worse biology, the following three analyses were performed. The immune cytolytic activity (CYT) score is a new indicator of cancer immunity developed by Rooney et al. [13]. It is acquired by calculating mRNA expression levels of granzyme A (GZMA) and perforin (PRF1) in the tumor and used to predict the level of immune-mediated elimination associated with cytotoxic T cell (CTL) markers. We have previously demonstrated that CYT score reflects immune cell antitumor activity and its clinical relevance in several articles [17,18,22,24,26-29,31,35-37,41,43,44,46-49,52-54].

Intratumor heterogeneity is the clonal variation between individual cancer cells within a patient's tumor. It has recently drawn attention as an important underlying mechanism of therapeutic failure. We utilized Mutant Allele Tumor Heterogeneity (MATH) to assess intratumor heterogeneity based on the exome sequencing of tumors [14]. Our prior work assessed intratumor heterogeneity and demonstrated that it is associated with cancer biology and patient outcomes [30,36,37,41,42,55].

Recently, tumor immune microenvironment has been found to plays an important role in cancer biology, and the profiling of the infiltrating immune cells are critical to assess the characteristics of a tumor [56]. CIBERSORT deconvolution algorithm was utilized by Newman et al. to estimate tumor-infiltrating immune cells in the tumor microenvironment using gene expression profiles [15]. The twenty-two cell fractions were calculated via their online calculator (https://cibersort.stanford.edu/). We have already illustrated that CIBERSORT analysis could provide the clinical significance and the prediction of outcomes in some cancers in several articles [21,27-31,35,36,41,48,52,57].

Comment 3:

the last 4 lines of discussions need to be more extensive. They should explain that menopause by itself can have major impact on biology, and that it would have been interesting to investigate the differences between premenopausal, postmenopausal <70, and older pts

 Response 3:

We totally agree with the Reviewer that menopause can have major impact on biology, thus comparison by menstruation status is of interest. We added the sentence in the discussion section as below.

Discussion section:

Also, it would have been ideal to group the patients by menopausal status; however, that information was lacking in both of the analyzed cohorts, and the sample size would be too small for meaningful analyses if we limited the control group to age 55-65.

Menopause significantly changes the hormonal environment of women and causes various symptoms [80]. It is well known that menopause by itself have major impact on breast cancer biology. Medication for hormone-sensitive breast cancer differs in pre-menopause patients and in post-menopause patients [81]. It has also been reported that zoledronic acid have different therapeutic effects before and after menopause [82]. However, few studies have compared the cancer biology by the menstruation status instead of age. To this end, it would be interesting to investigate the difference between premenopausal, postmenopausal under 70 years old, and older patients, since we were unable to do this with lack of menstruation information in our cohorts. By comparing the biology of pre- and post-menopausal cancer at the genetic level and clarifying the characteristics of each, it could be possible to contribute to the prevention and the treatment of breast cancer.

 Comment 2:

The methods section should more clearly contain the research questions they had prior to starting the study. Now they just mention the tools they used -

 Response 2:

We agree with the Reviewer that the method section should contain the research question more clearly. We added some sentences in the method section as below.

 2.1. Breast Cancer Patient Cohorts

Two completely independent breast cancer cohorts, The Cancer Genome Atlas (TCGA) and the Molecular Taxonomy of Breast Cancer International Consortium (METABRIC) were used in the current study. It is essential to analyze the cohorts with reasonable number of Octogenarian breast cancer patients to obtain statistically meaningful results, and these were the largest cohorts that we had access to. TCGA is a joint collaboration project of the National Cancer Institute (NCI) and the National Human Genome Research Institute of the National Institute of Health that includes 1084 breast cancer patients [33]. The METABRIC cohort was collected from tumor banks in the UK and Canada and includes 2509 breast cancer patients [34]. Both cohorts consist of treatment naïve primary cancer samples with clinical profiles, survival data, and gene expression data. They are de-identified publicly available databases; thus, institutional review board (IRB) approval was waived. Our group has used both of these databases previously to demonstrate and unravel the tumor biology of breast cancer [16-31,35-49].

 2.2. Statistical Analysis

In order to compare the biology between super-old and the most common age breast cancer patients, this study defined the control group as patients aged 40-65 and octogenarians (age over 80) at the time of breast cancer diagnosis. We used the Kaplan–Meier method with log-rank testing Two cohort has the information of disease specific survival, which is considered only breast cancer death, so we used the information as breast cancer specific survival following DATECAN guide-lines [50]. Fisher's exact tests and one-way ANOVA tests were used to compare the differences between groups. In all analyses, p-values less than 0.05 were considered statistically significant. All statistical analyses were performed using Microsoft Excel 2010, R software (http:///www.rproject.org/), and Bioconductor (http://bioconductor.org/).

 2.3. Gene Set Enrichment Analysis (GSEA)

Gene Set Enrichment Analysis (GSEA) was conducted to investigate what biology are associated with Octogenarian breast cancer compared to common age. GSEA was calculated by the Broad Institute software (http://software.broadinstitute.org/gsea/index.jsp) [51], as previously de-scribed [35,36]. As recommended by GSEA software, a False Positive Rate (FDR) of 0.25 was defined as statistically significant.

 2.4. The Immune Cytolytic Activity (CYT) Score, the Mutant-Allele Tumor Heterogeneity (MATH), and CIBERSORT

To substantiate our hypothesis that Octogenarian breast cancer has worse biology, the following three analyses were performed. The immune cytolytic activity (CYT) score is a new indicator of cancer immunity developed by Rooney et al. [13]. It is acquired by calculating mRNA expression levels of granzyme A (GZMA) and perforin (PRF1) in the tumor and used to predict the level of immune-mediated elimination associated with cytotoxic T cell (CTL) markers. We have previously demonstrated that CYT score reflects immune cell antitumor activity and its clinical relevance in several articles [17,18,22,24,26-29,31,35-37,41,43,44,46-49,52-54].

Intratumor heterogeneity is the clonal variation between individual cancer cells within a patient's tumor. It has recently drawn attention as an important underlying mechanism of therapeutic failure. We utilized Mutant Allele Tumor Heterogeneity (MATH) to assess intratumor heterogeneity based on the exome sequencing of tumors [14]. Our prior work assessed intratumor heterogeneity and demonstrated that it is associated with cancer biology and patient outcomes [30,36,37,41,42,55].

Recently, tumor immune microenvironment has been found to plays an important role in cancer biology, and the profiling of the infiltrating immune cells are critical to assess the characteristics of a tumor [56]. CIBERSORT deconvolution algorithm was utilized by Newman et al. to estimate tumor-infiltrating immune cells in the tumor microenvironment using gene expression profiles [15]. The twenty-two cell fractions were calculated via their online calculator (https://cibersort.stanford.edu/). We have already illustrated that CIBERSORT analysis could provide the clinical significance and the prediction of outcomes in some cancers in several articles [21,27-31,35,36,41,48,52,57].

Comment 3:

the last 4 lines of discussions need to be more extensive. They should explain that menopause by itself can have major impact on biology, and that it would have been interesting to investigate the differences between premenopausal, postmenopausal <70, and older pts

 Response 3:

We totally agree with the Reviewer that menopause can have major impact on biology, thus comparison by menstruation status is of interest. We added the sentence in the discussion section as below.

Discussion section:

Also, it would have been ideal to group the patients by menopausal status; however, that information was lacking in both of the analyzed cohorts, and the sample size would be too small for meaningful analyses if we limited the control group to age 55-65.

Menopause significantly changes the hormonal environment of women and causes various symptoms [80]. It is well known that menopause by itself have major impact on breast cancer biology. Medication for hormone-sensitive breast cancer differs in pre-menopause patients and in post-menopause patients [81]. It has also been reported that zoledronic acid have different therapeutic effects before and after menopause [82]. However, few studies have compared the cancer biology by the menstruation status instead of age. To this end, it would be interesting to investigate the difference between premenopausal, postmenopausal under 70 years old, and older patients, since we were unable to do this with lack of menstruation information in our cohorts. By comparing the biology of pre- and post-menopausal cancer at the genetic level and clarifying the characteristics of each, it could be possible to contribute to the prevention and the treatment of breast cancer.

Comment 2:

The methods section should more clearly contain the research questions they had prior to starting the study. Now they just mention the tools they used -

 Response 2:

We agree with the Reviewer that the method section should contain the research question more clearly. We added some sentences in the method section as below.

 2.1. Breast Cancer Patient Cohorts

Two completely independent breast cancer cohorts, The Cancer Genome Atlas (TCGA) and the Molecular Taxonomy of Breast Cancer International Consortium (METABRIC) were used in the current study. It is essential to analyze the cohorts with reasonable number of Octogenarian breast cancer patients to obtain statistically meaningful results, and these were the largest cohorts that we had access to. TCGA is a joint collaboration project of the National Cancer Institute (NCI) and the National Human Genome Research Institute of the National Institute of Health that includes 1084 breast cancer patients [33]. The METABRIC cohort was collected from tumor banks in the UK and Canada and includes 2509 breast cancer patients [34]. Both cohorts consist of treatment naïve primary cancer samples with clinical profiles, survival data, and gene expression data. They are de-identified publicly available databases; thus, institutional review board (IRB) approval was waived. Our group has used both of these databases previously to demonstrate and unravel the tumor biology of breast cancer [16-31,35-49].

 2.2. Statistical Analysis

In order to compare the biology between super-old and the most common age breast cancer patients, this study defined the control group as patients aged 40-65 and octogenarians (age over 80) at the time of breast cancer diagnosis. We used the Kaplan–Meier method with log-rank testing Two cohort has the information of disease specific survival, which is considered only breast cancer death, so we used the information as breast cancer specific survival following DATECAN guide-lines [50]. Fisher's exact tests and one-way ANOVA tests were used to compare the differences between groups. In all analyses, p-values less than 0.05 were considered statistically significant. All statistical analyses were performed using Microsoft Excel 2010, R software (http:///www.rproject.org/), and Bioconductor (http://bioconductor.org/).

 2.3. Gene Set Enrichment Analysis (GSEA)

Gene Set Enrichment Analysis (GSEA) was conducted to investigate what biology are associated with Octogenarian breast cancer compared to common age. GSEA was calculated by the Broad Institute software (http://software.broadinstitute.org/gsea/index.jsp) [51], as previously de-scribed [35,36]. As recommended by GSEA software, a False Positive Rate (FDR) of 0.25 was defined as statistically significant.

 2.4. The Immune Cytolytic Activity (CYT) Score, the Mutant-Allele Tumor Heterogeneity (MATH), and CIBERSORT

To substantiate our hypothesis that Octogenarian breast cancer has worse biology, the following three analyses were performed. The immune cytolytic activity (CYT) score is a new indicator of cancer immunity developed by Rooney et al. [13]. It is acquired by calculating mRNA expression levels of granzyme A (GZMA) and perforin (PRF1) in the tumor and used to predict the level of immune-mediated elimination associated with cytotoxic T cell (CTL) markers. We have previously demonstrated that CYT score reflects immune cell antitumor activity and its clinical relevance in several articles [17,18,22,24,26-29,31,35-37,41,43,44,46-49,52-54].

Intratumor heterogeneity is the clonal variation between individual cancer cells within a patient's tumor. It has recently drawn attention as an important underlying mechanism of therapeutic failure. We utilized Mutant Allele Tumor Heterogeneity (MATH) to assess intratumor heterogeneity based on the exome sequencing of tumors [14]. Our prior work assessed intratumor heterogeneity and demonstrated that it is associated with cancer biology and patient outcomes [30,36,37,41,42,55].

Recently, tumor immune microenvironment has been found to plays an important role in cancer biology, and the profiling of the infiltrating immune cells are critical to assess the characteristics of a tumor [56]. CIBERSORT deconvolution algorithm was utilized by Newman et al. to estimate tumor-infiltrating immune cells in the tumor microenvironment using gene expression profiles [15]. The twenty-two cell fractions were calculated via their online calculator (https://cibersort.stanford.edu/). We have already illustrated that CIBERSORT analysis could provide the clinical significance and the prediction of outcomes in some cancers in several articles [21,27-31,35,36,41,48,52,57].

Comment 3:

the last 4 lines of discussions need to be more extensive. They should explain that menopause by itself can have major impact on biology, and that it would have been interesting to investigate the differences between premenopausal, postmenopausal <70, and older pts

 Response 3:

We totally agree with the Reviewer that menopause can have major impact on biology, thus comparison by menstruation status is of interest. We added the sentence in the discussion section as below.

Discussion section:

Also, it would have been ideal to group the patients by menopausal status; however, that information was lacking in both of the analyzed cohorts, and the sample size would be too small for meaningful analyses if we limited the control group to age 55-65.

Menopause significantly changes the hormonal environment of women and causes various symptoms [80]. It is well known that menopause by itself have major impact on breast cancer biology. Medication for hormone-sensitive breast cancer differs in pre-menopause patients and in post-menopause patients [81]. It has also been reported that zoledronic acid have different therapeutic effects before and after menopause [82]. However, few studies have compared the cancer biology by the menstruation status instead of age. To this end, it would be interesting to investigate the difference between premenopausal, postmenopausal under 70 years old, and older patients, since we were unable to do this with lack of menstruation information in our cohorts. By comparing the biology of pre- and post-menopausal cancer at the genetic level and clarifying the characteristics of each, it could be possible to contribute to the prevention and the treatment of breast cancer.

Round 3

Reviewer 3 Report

ok, they included all the previous remarks